# Genetic Mutations and Mitochondrial Redox Signaling as Modulating Factors in Hypertrophic Cardiomyopathy: A Scoping Review

**DOI:** 10.3390/ijms25115855

**Published:** 2024-05-28

**Authors:** Antonio da Silva Menezes Junior, Ana Luísa Guedes de França-e-Silva, Henrique Lima de Oliveira, Khissya Beatryz Alves de Lima, Iane de Oliveira Pires Porto, Thays Millena Alves Pedroso, Daniela de Melo e Silva, Aguinaldo F. Freitas

**Affiliations:** 1Faculdade de Medicina, Departamento de Clínica Médica, Universidade Federal de Goiás (UFG), Goiânia 74020-020, Brazil; ana.guedes@discente.ufg.br (A.L.G.d.F.-e.-S.); henrique.lima2@discente.ufg.br (H.L.d.O.); khissya_beatryz@discente.ufg.br (K.B.A.d.L.); danielamelosilva@ufg.br (D.d.M.e.S.); afreitasjr@msn.com (A.F.F.J.); 2Faculdade de Medicina, Universidade de Rio Verde (UniRV), Campus Aparecida, Aparecida de Goiânia 74345-030, Brazil; iane.porto@unirv.edu.br (I.d.O.P.P.); thays.millena04@gmail.com (T.M.A.P.)

**Keywords:** hypertrophic cardiomyopathy, metabolism, mitochondrial redox signaling, mitochondrial dysfunction

## Abstract

Hypertrophic cardiomyopathy (HCM) is a heart condition characterized by cellular and metabolic dysfunction, with mitochondrial dysfunction playing a crucial role. Although the direct relationship between genetic mutations and mitochondrial dysfunction remains unclear, targeting mitochondrial dysfunction presents promising opportunities for treatment, as there are currently no effective treatments available for HCM. This review adhered to the Preferred Reporting Items for Systematic Reviews and Meta-Analysis Extension for Scoping Reviews guidelines. Searches were conducted in databases such as PubMed, Embase, and Scopus up to September 2023 using “MESH terms”. Bibliographic references from pertinent articles were also included. Hypertrophic cardiomyopathy (HCM) is influenced by ionic homeostasis, cardiac tissue remodeling, metabolic balance, genetic mutations, reactive oxygen species regulation, and mitochondrial dysfunction. The latter is a common factor regardless of the cause and is linked to intracellular calcium handling, energetic and oxidative stress, and HCM-induced hypertrophy. Hypertrophic cardiomyopathy treatments focus on symptom management and complication prevention. Targeted therapeutic approaches, such as improving mitochondrial bioenergetics, are being explored. This includes coenzyme Q and elamipretide therapies and metabolic strategies like therapeutic ketosis. Understanding the biomolecular, genetic, and mitochondrial mechanisms underlying HCM is crucial for developing new therapeutic modalities.

## 1. Introduction

Hypertrophic cardiomyopathy (HCM) results from microcellular changes such as myocyte remodeling, disorganization of sarcomeric proteins, and energy metabolism impairment. This condition clinically manifests itself in left ventricular hypertrophy and abnormal diastolic function. HCM is rare, with a prevalence rate of 0.2–0.5% in adults, and is a significant cause of sudden cardiac death [1,2,3,4,5]. By understanding the underlying causes of HCM, medical professionals can better diagnose and treat the condition, improving patient outcomes [1,2,3].

The causes of HCM are multifaceted. Genetic mutations in sarcomeric proteins account for 40–60% of cases, while intrinsic mutations in mitochondrial DNA, inborn metabolic errors, and other genetic and non-genetic factors also play a role [2,6,7]. Mitochondrial dysfunction is consistently present in HCM pathophysiology, regardless of the underlying cause [8]. This cascade of molecular disturbances affects intracellular calcium homeostasis, cellular metabolism, and the production of reactive oxygen species, including superoxide anions, hydroxyl, and peroxyl radicals [9,10,11]. It is important to understand that the illness is not a straightforward condition. The various changes collectively contribute to the complexity of the illness, making it a challenge to treat. Without a comprehensive understanding of these changes, it is not easy to develop effective treatment options. Therefore, we must continue to study and learn more about the pathophysiological complexity of the illness to provide better care for those affected by it [12,13,14,15,16,17,18,19,20,21,22,23].

Genetic variations, mitochondrial dysfunction, and oxidative stress are significant factors in the development and pathological remodeling of hypertrophic cardiomyopathy (HCM). The impact of these factors is evident in the changed structure of heart cells, mitochondrial cristae, and the functioning of oxidative phosphorylation complexes [8,12,13,14,15,16,17,18,19,20,21,22,23]. Currently, our understanding of the genetic mutations that cause HCM is limited. As a result, the relationship between these mutations, mitochondrial modulation, and HCM is not yet fully understood [19,24]. Mitochondrial dysfunction can be either primary or secondary to other etiologies [23,25].

Mitochondrial dysfunction is associated with sarcomeric mutations that can significantly impact the tertiary structure and stability of proteins [23,26]. Inefficient sarcomere performance increases ATP demand, leading to energy depletion, chronic workload increases, and oxidative stress [27]. It is essential to address the contributing factors proactively to prevent further complications down the line, as early manifestations of metabolic disorders in HCM progression are a direct result of them. It is clear that understanding and preventing these mutations is key; therefore, preventing or ameliorating mitochondrial dysfunction by targeting metabolic and oxidative stress may offer viable therapeutic strategies for halting or reversing disease progression [15,16,20,23].

HCM is a condition that can be quite debilitating and currently has no known cure; however, mavacamten, a promising ATPase inhibitor, has been approved by the FDA and can help reduce the need for invasive procedures in individuals with the obstructive form of HCM [28]. Although mavacamten cannot cure HCM, it has demonstrated a considerable potential to reduce cardiac contractility and alleviate energy depletion and oxidative stress. To develop effective treatments, it is crucial to comprehend the initial pathophysiological mechanisms of the condition, and researchers have analyzed various variants of the pathophysiology of HCM in detail [8]. Mitochondrial and genetic therapies are emerging potential methods for preventing and improving heart disease in HCM [23,29]. With further exploration of treatment options and ongoing research into the disease’s pathophysiology, we can discover alternative treatments for HCM and enhance the quality of life for those with it.

## 2. Materials and Methods

This review adhered to the Preferred Reporting Items for Systematic Reviews and Meta-Analysis Extension for Scoping Reviews (PRISMA-ScR) guidelines. Searches were conducted in databases such as PubMed, EMBASE, and Scopus up to February 2024 using “MESH terms”. Bibliographic references from pertinent articles were also included.

### 2.1. Protocol and Registration

This review has been registered with the Open Science Framework (OSF) and is accessible at https://doi.org/10.17605/OSF.IO/XJGVC since 16 October 2023.

### 2.2. Eligibility Criteria

Our review focused on articles that connected metabolic dysfunction, sarcomeric or DNA mitochondrial mutations, G-negatives, mitochondrial architecture, and functioning disorders with HCM. We included articles from peer-reviewed journals published until February 2024 that discussed HCM in the context of mitochondrial dysfunction and related mutations. The scope of our review encompasses quantitative, qualitative, and mixed-methods studies addressing genetic mutations and mitochondrial dysfunction as factors in HCM. Articles exclusively focused on heart or mitochondrial diseases unrelated to HCM were excluded. However, articles elucidating the role of calcium in metabolic changes associated with HCM were included.

### 2.3. Exclusion Criteria

Excluded from consideration in the review were secondary sources such as editorials, books, expert opinion articles, dissertations, theses, and conference abstracts, except for literature reviews, which were included.

### 2.4. Data Sources

We conducted comprehensive searches of the Excerpta Medica Database (Embase), SciVerse Scopus, and PubMed databases. Our search strategy involved a combination of controlled descriptors and keywords pertinent to HCM, and no restrictions were imposed on the language or publication period. In addition, reference lists of the initially selected studies were manually searched to identify other relevant articles.

### 2.5. Study Selection Process

The identified studies were imported into the Rayyan software [30] (https://link.springer.com/article/10.1186/s13643-016-0384-4 accessed on 22 April 2024) for duplicate removal and evaluation of the eligibility criteria. This process involved three independent blinded reviewers (ASMJr, ALGFS, and HLO) who initially examined the titles and abstracts (Phase 1), followed by a full-text review of the studies selected in Phase 1 (Phase 2). A fourth reviewer resolved any discrepancy in the selection process.

### 2.6. Data Extraction Process

Data extraction was conducted by three independent blinded reviewers using a characterization table created using Microsoft Word (2021). This table includes study characteristics, including identification (citation), study design, and country; analyzed characteristics of HCM, including etiology, sample size, sex, and average age; and main outcomes, including molecular manifestations such as mitochondrial dysfunction, oxidative stress, and metabolic disorders.

### 2.7. Data Synthesis

Data from the selected studies were qualitatively synthesized, focusing on the proposed genetic mutations and mitochondrial dysfunction. This synthesis includes consideration of the etiology of HCM, whether from genetic mutations (sarcomeric or mitochondrial) and cellular metabolic dysfunction. All information was organized into a descriptive table for a comprehensive analysis.

## 3. Results and Discussion

Thirty-two articles were selected for the final analysis from an initial search of 597 (Figure 1). The primary data from these studies are presented in Appendix A. Our analysis focused on two primary categories: (1) Genetic Mutations in HCM; (2) Mitochondrial Redox Signaling.

### 3.1. Genetic Mutations in HCM

HCM is a complex disorder that results from a combination of genetic and non-genetic factors. (Figure 2) [2,31,32]. Additionally, metabolic alterations and mitochondrial dysfunction can give rise to this pathology as primary disorders or secondary conditions to the primary etiology. The cardiac beta-myosin heavy chain (MYH7) and myosin-binding protein C (MYBPC) genes are responsible for roughly 70% of identifiable mutations [33]. MYH7 mutations affect myosin ATPase activity, leading to increased myocardial force. In contrast, MYBPC mutations are involved in sarcomere organization and may regulate myofibril contraction, resulting in energy overload and hypercontractility in the left ventricle [34,35]. In addition to MYH7 and MYBPC3, other genes, such as TNNT2, TNNI3, TPM1, ACTC1, MYL2, MYL3, and CSRP3, are also implicated in HCM, albeit less frequently. Some pathogenic variants exhibit high penetrance and are causal mutations, while others demonstrate incomplete penetrance influenced by genetic and environmental factors.

In HCM, allelic imbalance, particularly the characteristic autosomal dominant inheritance, can result in mosaic expression patterns of the mutant proteins, leading to cellular variability and causing differences in calcium sensitivity and contractile force in the cardiac cells [29,36].

#### 3.1.1. Epigenetics and HCM

Epigenetic modifications play a significant role in HCM by influencing the levels of mutated proteins. Specifically, the DNA methylation in the promoter of the MYH7 gene is inversely correlated with beta-MyHC messenger RNA (mRNA) levels. This suggests that changes in methylation can alter the transcription of the mutated gene [26]. Moreover, the activation of chromatin-remodeling proteins, such as BRG1 and DPF3a, is associated with disease severity in patients with HCM [37]. These epigenetic changes impact the expression of mutated genes, fetal gene programs, and other hypertrophic genes [38].

These findings have significant implications for the diagnosis, treatment, and management of HCM. By understanding the role of epigenetic modifications in the expression of mutated genes, targeted therapies that address the underlying cause of the disease may be possible. Identifying specific chromatin-remodeling proteins associated with disease severity may also lead to the development of personalized treatment options that consider individual differences in genetic and epigenetic profiles.

As such, continued research into the relationship between epigenetic modifications and HCM is essential for advancing our understanding of this complex disease. By exploring the underlying mechanisms of epigenetic changes, we may be able to identify new targets for therapeutic intervention and improve the lives of individuals living with HCM.

#### 3.1.2. Myc Gene in HCM

The Myc gene encodes a vital transcription factor in regulating cellular proliferation and growth. It is also a key player in the development and progression of HCM. Animal models have consistently shown that overexpression of Myc leads to cell cycle activation and tumorigenesis [39]. Interestingly, elevated Myc mRNA levels have also been observed in mouse and patient samples with HCM [12,40]. Myc acts on mitochondrial function by modulating mitochondrial biogenesis through the PGC-1a pathway, which is critical in changing HCM and heart failure. PGC-1a, a master regulator, coordinates the expression of genes vital for mitochondrial formation and function and is involved in metabolic processes like glucose control, lipid metabolism, response to oxidative stress, and adaptation to various conditions. The regulation of PGC-1a is essential for overall metabolic health and cellular function, particularly in cardiac conditions such as cardiomyopathy [12].

#### 3.1.3. Mitochondrial DNA Mutations and HCM

Studies have shown that patients with sarcomere mutations may experience changes in heart efficiency even before hypertrophy development, indicating a link between HCM and metabolic alterations [23]. Moreover, point mutations in mitochondrial DNA (mtDNA), including m.3243A>G, m.3302A>G, m.4300A>G, and m.8344A>G, can impact ATP production and increase oxidative stress, ultimately contributing to HCM. The identification of the first mtDNA mutation related to HCM, MT-TL1, in 1991 led to significant strides in understanding the mechanisms of gene expression associated with these mutations, such as 1-deoxynojirimycin, which can aid in the recovery of mitochondrial cristae [41].

Mutations in mitochondrial genes such as MT-RNR2, ELAC2, Gtpbp3, and Mto1 have also been linked to HCM [14,16,18,20,42,43]. Research following the discovery of MT-TL1 mutations has expanded our understanding of mitochondrial gene mutations associated with HCM, revealing their impact on mitochondrial function and energy production in the pathological processes leading to HCM [41].

#### 3.1.4. MicroRNAs and Gene Therapy in HCM

Gene therapy emerges as an advanced medical therapy focusing on the genetic modification of cells for therapeutic purposes. In the context of HCM, it encompasses genome editing techniques, gene replacement therapy, and allele-specific silencing [44,45]. CRISPR-Cas9 technology presents promising potential to correct genetic mutations underlying HCM. It is at the forefront of human germline therapy, with several studies reporting efficient and successful genetic editing in these embryos [44,46]. However, it faces obstacles, including limited efficiency in homology-directed repair in somatic cardiac cells.

miRNAs represent potential targets in gene therapy to optimize metabolism and energy delivery in hypertrophied hearts. MicroRNA-146a inhibits oxidative metabolism, attenuates the hypertrophic response, and reduces cardiac erbB4 signaling, which regulates glucose metabolism [47,48]. RNA interference (RNAi) therapy and MYBPC3 cDNA replacement via gene transfer have shown therapeutic potential. However, allele-specific gene silencing with RNAi faces challenges such as off-target effects and reliance on adenoviruses [34]. Despite these hurdles, gene editing technology is emerging as a crucial future therapy [49,50]. Moreover, viral vectors, especially adeno-associated viruses (AAVs), notably AAV9, the most cardiotropic serotype, demonstrate efficacy in precisely delivering genetic material to cardiac tissue [45,51]. However, potential immune responses and ethical considerations pose considerable challenges. Furthermore, the MYBPC3 gene replacement approach is promising, particularly due to its ability to correct cMyBP-C haploinsufficiency, reduce hypertrophy, and maintain sarcomeric stoichiometry [52].

Understanding these advancements is a crucial starting point for direct interventions in HCM pathogenesis, offering promising prospects for a definitive approach to treating hereditary diseases. Ethical considerations, including germline gene editing, are thoroughly addressed to ensure the responsible application of these innovative technologies [45]. Technical challenges associated with efficient Cas9-gRNA delivery, such as immune responses to viral vectors and issues related to non-viral vector delivery, can be overcome. In that case, CRISPR-Cas9 positions itself as a significant player in treating a broad range of disorders where partial or complete gene elimination is desired [53]. This potentially revolutionary scenario instills hope for substantial clinical advancements in the coming years, notwithstanding concerns such as mosaicism, off-target alterations, and, ultimately, non-Mendelian cases of HCM, which must be meticulously considered.

### 3.2. Mitochondrial Redox Signaling

Several studies have reported reduced ATP production due to impaired mitochondrial metabolism, as shown in Figure 3 [8,12,13,14,15,16,17,18,19,20,21,22,23,29]. Adenosine triphosphate (ATP) production in a healthy human heart relies primarily on fatty acids and glucose oxidation. However, when the cardiac workload increases, the ATP generation pathway shifts toward glucose oxidation. This shift is regulated by the Randle cycle, which facilitates the maintenance of energy balance and increased energy efficiency in the heart [9,15,19].

In hearts affected by HCM, however, the increased energy demand during heart contraction can lead to metabolic disorders characterized by reduced ATP levels and increased intracellular ADP [26,54], with a preference for mitochondrial substrates over glucose [45]. This metabolic shift is also observed in patients with chronic myocardial infarction (CMI). It leads to cardiac hypertrophy due to increased ATP uptake, resulting in the accumulation of fatty acids within cells, causing lipotoxicity rather than increased glucose oxidation [3,9,29,55]. This metabolic deviation is a distinct characteristic of HCM, occurring independently of the sarcomeric protein genotype. HCM also reduces capillary density in cardiac tissue and hypoxia, prompting a metabolic shift from aerobic glucose metabolism to anaerobic glycolysis. This shift reduces energy efficiency and contributes to heart hypertrophy in a self-perpetuating cycle [15]. Metabolomic analyses have shown reduced cardiac bioenergetics and substrate utilization, indicating disturbances in energy metabolism that may contribute to the development of HCM-associated pathology [19].

A recent multi-omics study reported decreased metabolic supplementation through the TCA/glutamine pathway and changes in the pentose phosphate pathway. These pathways are fundamental for NADPH production and antioxidant protection. Therefore, comprehending the alterations in energy metabolism and metabolic pathways can help identify therapeutic targets for clinical improvement in HCM [22]. The therapeutic potential of perhexylin treatment, which shifts the heart’s substrate utilization from fatty acids to glucose, has been demonstrated in patients with HCM. This treatment also functions as a NOX2 inhibitor, reducing oxidative stress in the heart [29,56]. Notably, these metabolic changes are partially reversible, and early detection can offer therapeutic opportunities.

Reduced mitochondrial respiration linked to NADH oxidation is a significant functional impairment in hypertrophic cardiomyopathy (HCM) [23]. However, mutations in both sarcomeric and non-sarcomeric genes, such as those in CSRP3, can potentially disturb calcium homeostasis [8]. The Muscle LIM Protein (MLP) maintains intracellular calcium balance and contributes to its pathogenesis [8,15,22]. Furthermore, the CACNA1C gene encodes the pore-forming subunit of the L-type calcium channel CaV1.2 [57]. Increased transcriptional expression of CACNA1C enhances calcium release from the sarcoplasmic reticulum (SR) and diminishes cytoplasmic calcium transport back to the SR during diastole [58,59]. These factors hinder sarcomeric relaxation, leading to diastolic dysfunction and subsequent adverse effects, including mitochondrial dysfunction, oxidative stress, and alterations in calcium-dependent pathways [17,60].

Moreover, MLP mutations raise energy demands due to inefficient sarcomeric ATP use. This leads to increased reactive oxygen species generation and further progression of HCM and heart failure. The MT-RNR2 mutation affects mitochondrial DNA, causing dysfunctions such as a decreased ATP/ADP ratio and compromised mitochondrial membrane integrity [59]. These alterations elevate intracellular Ca^2+^ concentration, thereby affecting intracellular homeostasis. Research on pluripotent stem cells has shown reduced mitochondrial calcium uniporter protein levels and increased intracellular Ca^2+^ concentrations and SR Ca^2+^ reserves. The lowered membrane potential reduces Ca^2+^ intake, potentially disrupting calcium homeostasis and triggering calcium-dependent signaling pathways contributing to heart hypertrophy [14]. This mutation also increases the number of mitochondria, possibly as an adaptation to decreased energy generation. Mutations in the *MYBPC3* and *MYH7* genes have increased calcium sensitivity in cardiac myofilaments, resulting in compromised contractility and increased ATP consumption. As a result, this hurts HCM by disturbing the balance of calcium levels and metabolic processes. Several investigations, including those undertaken by [2,41,54,59,61], have been conducted.

The binding of calcium to myofilaments leads to a decline in the regulation of cytosolic Ca^2+^, inhibiting the uptake of mitochondrial Ca^2+^. This process negatively affects the Krebs cycle-mediated replenishment of reduced NADH and NADPH, reducing the overall rate of metabolic reactions and modulating the system’s efficiency [62]. In situations with a high demand for ATP, ADP levels increase, and NADH and NADPH levels decrease, leading to an imbalance, as shown in Figure 4, results in oxidative stress due to heightened levels of ADP that exacerbate NADH and NADPH oxidation. This disruption of the equilibrium between NAD and its oxidized form, NAD^+^, is crucial for cellular energy production, leading to increased ATP usage. This process is associated with the initial inefficient performance of mutation-induced sarcomeres, which leads to chronic cardiac workload, energy deficiency, and oxidative stress [15,54,63].

Elevated levels of ADP have been shown to increase the sensitivity and affinity of calcium in myofibrils, thereby contributing to heart dysfunction. In normal cardiomyocytes, the enzyme creatine kinase is essential in regenerating ATP in myofilaments using mitochondrial phosphocreatine (PCr). However, in cardiomyocytes suffering from heart failure, creatine kinase does not succeed in effectively reducing ADP concentrations. This failure may contribute to the altered calcium sensitivity and affinity observed in such cells, exacerbating heart failure [64]. Elevated ADP levels and reduced ATP impair ATP-dependent ion pumps in the heart, specifically SERCA, which recaptures calcium into SR [8,65,66]. According to Wijnker et al., the direct and indirect effects of various factors significantly impact the development of HCM [29].

Patients with sarcomeric mutations often exhibit increased septum thickness, which is connected to improved mitochondrial respiration using succinate. However, they also experience reduced cellular respiration involving NADH, as per a study conducted by Lucas et al. (2003) and Nollet et al. (2023a) [23,63]. Nollet et al. (2023) further identified defects in NAD^+^ homeostasis in myectomy samples of patients with HCM, which led to compromised bioenergetics due to diminished NADH conversion. This conversion process, which converts NADH (nicotinamide adenine dinucleotide reduced) into NAD+ (nicotinamide adenine dinucleotide), is crucial for the transfer of electrons during cellular energy production, as explained by Farhana et al. (2023) [67]. The study also revealed no incorporation of complex I in respiratory super complexes and peroxidation of cardiolipin, destabilizing the mitochondrial membrane. The use of cardiolipin-stabilizing compounds for treatment could potentially enhance the mitochondrial respiration linked to NADH [23].

Mitochondrial dysfunction is a key driver of pathological remodeling in HCM, particularly in genotype-negative patients with impaired NADH-related respiration. These early metabolic changes suggest that addressing metabolic stress could be a strategic approach to treating and potentially reversing its progression [23,63,68]. Ranjbarvaziri et al. (2021) demonstrated significant energy depletion in genetically linked patients with HCM, characterized by reduced high-energy phosphate metabolites and decreased mitochondrial genes involved in creatine kinase and ATP synthesis [19]. This energy deficit activates the AMPK pathway, a cellular stress response sensor with low ATP levels [19].

Significant progress has been made in understanding the relationship between structural, sarcomeric, and mitochondrial functions in HCM. However, more research is still needed to fully comprehend the mechanisms that cause changes in cardiac energy utilization in the different HCM phenotypes. According to a study by Viola et al. (2019), specific mutations cause distinct changes in intracellular calcium homeostasis and mitochondrial metabolic function [8]. This highlights the importance of understanding the pathophysiological mechanisms of disease mutations to develop effective drug therapies. Lee et al. (2009) found a connection between mitochondrial respiration dysfunction and other disorders in transgenic mice overexpressing Myc oncogene [12]. These dysfunctions can lead to lower production of energy and an increased formation of oxygen-free radicals, which can worsen mitochondrial damage [39]. The expression of Myc oncogene in patients with terminal heart failure provides new possibilities for therapeutic approaches in the treatment and management of the condition [12].

Cardiomyocytes undergo oxidative phosphorylation, producing reactive oxygen species (ROS) in the mitochondria. The respiratory chain generates most ROS, with complexes I and III being the primary sources, as shown in Figure 5. Superoxide anion is produced in complex I through reduced flavin mononucleotide or the iron-sulfur clusters N-1a and N-1b, while in complex III, it is generated at the ubiquinol oxidation site. The respiratory chain has cellular antioxidant mechanisms that activate when an accidental leakage of electrons to oxygen occurs during energy production. These ROS, including superoxide anion, hydroxyl radical, and hydrogen peroxide, are critical signaling molecules that play crucial roles in cardiac physiology and disease [13,29,59]. Cytosolic and mitochondrial sources of ROS contribute to the intracellular pool. Under normal conditions, ROS signaling regulates cardiac development and cardiomyocyte maturation, cardiac calcium handling, excitation-contraction coupling, and vascular tone [69]. Elevated ROS levels can cause oxidative damage to mitochondria and DNA, harm proteins and lipids, activate the mitochondrial permeability transition pore, cause mitochondrial dysfunction, and lead to cellular death. Additionally, it can disrupt the ATP and ADP balance [13,29,70].

In addition to the respiratory chain, several proteins in the mitochondria contribute to the mitochondrial pool of ROS. P66shc, located in the mitochondrial intermembrane space, plays a significant role in oxidative stress signaling. It contributes to mitochondrial ROS production by oxidizing cytochrome c and stimulating hydrogen peroxide production. Isoforms A and B of monoamine oxidase (MAO-A and MAO-B) are crucial sources of mitochondrial hydrogen peroxide. Located in the mitochondrial outer membrane, MAOs degrade monoamines into hydrogen peroxide and aldehydes [71]. Furthermore, NOXs are a family of proteins involved in intracellular ROS production [72]. NOX4 is partially located in the mitochondrial inner membrane and is a source of mitochondrial superoxide and hydrogen peroxide, whose activity is regulated by ATP [73].

While the exact mechanisms of HCM are still not fully understood, previous non-invasive studies have shed light on certain aspects of the disease. According to these studies, compromised energy metabolism during the early stages of HCM can contribute to disease progression. In particular, mitochondrial defects have been identified as a key factor in this process [3,54,74].

Chronic mitochondrial dysfunction in myocardial heart disease (MHD) can reduce the activity of complex I, which is the primary entry point for electrons in the mitochondrial respiratory chain. This impaired function can intensify oxidative stress and increase the release of ROS, which in turn stimulates cardiac hypertrophy [75]. Metabolomic analyses have revealed a decrease in the ratio of mitochondrial DNA to genomic DNA, reduced expression of genes linked to mtDNA integrity, mitochondrial transcription, translation, and diminished cardiolipin species. These factors can affect mitochondrial respiratory activity and membrane integrity [2,3,19,54].

An insight modeling study has indicated that hearts affected by HCM undergo electrophysiological changes due to altered mitochondrial membrane potential. This implies reduced oxygen radical production in the electron transport chain and subsequent energy expenditure beyond supply. This discrepancy can be attributed to the increased oxygen consumption of cardiomyocytes [76].

Mitochondrial ROS are a significant factor in the pathogenesis of several diseases, including HCM. Superoxide dismutases (SODs) are the primary defense against mitochondrial ROS. These enzymes convert superoxide into hydrogen peroxide, and three isoforms of SOD (SOD1, SOD2, and SOD3) are present in different cellular compartments, regulating specific pools of ROS. SOD1 is mainly found in the cytosol but also localizes to the mitochondrial intermembrane space. Conversely, SOD2 is situated within the mitochondrial matrix, and SOD3 is extracellular. Precise control over the localization and activity of SOD1 and SOD2 is vital to mitigate mitochondrial ROS [69,77].

In addition to SODs, other enzymes play crucial roles in hydrogen peroxide detoxification. Catalase, predominantly found in peroxisomes and potentially in cardiac mitochondria, catalyzes the breakdown of hydrogen peroxide into water and oxygen. GSH-PXs 1 and 4, situated within the mitochondria, utilize reduced glutathione (GSH) to convert hydrogen peroxide into water. PRXs, including PRX3 and PRX5 within the mitochondria, scavenge hydrogen peroxide and peroxynitrite. These enzymes are crucial for neutralizing ROS and replenishing antioxidants [78,79].

In HCM, elevated oxidized cysteine (cystine) levels and a high oxidized glutathione (GSSG) ratio to GSH indicate increased oxidative stress. Moreover, reduced levels of essential antioxidants such as superoxide dismutase, catalase, glutathione reductase (GSR), and glutathione peroxidase (GPX) suggest potential transcriptional alterations that could compromise overall antioxidant capacity [19].

It has been observed that understanding the important role of SODs and other enzymes in regulating ROS can provide significant insights into the pathogenesis of HCM. Precise control of the localization and activity of these enzymes is crucial to reducing mitochondrial ROS and replenishing antioxidant levels. Wijnker et al. (2019) suggested that antioxidant protection through thiol-based systems and the pentose phosphate pathway occurs during the early stages of HCM [29]. However, these protective mechanisms decrease in advanced stages, resulting in chronic oxidative stress, which contributes to HCM progression and creates a cycle of mitochondrial damage and ROS generation [3,29]. Early detection of oxidative stress is essential for controlling HCM progression, enabling effective interventions, and identifying potential therapeutic targets. Primary mitochondrial diseases, although rare, often result in oxidative phosphorylation dysfunction due to mtDNA variations, with approximately 40% of affected children developing heart diseases, including heart complications [80].

Mitochondrial cristae, which are critical for cell energy production, are significantly damaged by ROS. This leads to lipid and protein oxidation, lipid peroxidation, and protein modification, affecting mitochondrial DNA, causing mutations and impaired electron transport. Oxidative stress also triggers apoptosis and permeabilization of the inner mitochondrial membrane, contributing to the pathology of HCM [2,3,13,29].

Studies have shown significant changes in mitochondrial morphology in patients with HCM, including damaged structures with disorganized and less dense cristae and an increased number of mitochondria, indicative of mitochondrial division. However, Ranjbarvaziri et al. (2019) found no evidence of increased mitochondrial fission or fusion [19]. Nollet et al. (2022 and 2023a) linked mitochondrial dysfunction in HCM to inadequate organization of interfibrillar mitochondria characterized by isolated clusters disconnected from myofilaments [23,81]. This disorganization may impair mitochondrial bioenergetics, defense against ROS, and calcium cycle disruption, leading to global energetic and metabolic cellular dysfunction [23,81]. Furthermore, prominent changes in myocardial ultrastructure have been observed, including alterations in cellular organization, a significant increase in extracellular matrix formation, and a decrease in mitochondrial mass in the sub-sarcomeric portion of cardiomyocytes [68].

Mitophagy is crucial in maintaining mitochondrial integrity and function in the myocardium by eliminating the damaged mitochondria [82]. Dysfunctional mitochondria observed in clinical models and in vitro cultures of HCM indicate potential dysfunction of overwhelming mitophagy pathways. Significant downregulation of mitophagy-related genes has been reported in patients with HCM [19]. Findings by Nollet et al. (2023a) [23] suggest that deterioration of mitochondrial quality control mechanisms contributes to the accumulation of damaged mitochondria in HCM, exacerbating the mitochondrial injury and continuing a detrimental cycle in its progression [1,22,80,83].

#### 3.2.1. Emerging Therapeutic Strategies

This scoping review highlights the roles of genetic mutations and mitochondrial dysfunction in the development and progression of HCM. Metabolic modifications of HCM, characterized by an altered metabolic profile, abnormal mitochondrial structure, impaired respiratory function, and failure to regulate mitophagy, could guide the development of new therapeutic approaches to restore metabolic function and preserve mitochondrial integrity in the early stages of the disease. Studies have emphasized that cellular oxidative stress is a crucial factor in mitochondrial transformations, leading to structural and functional changes, alterations in cell signaling pathways, and uncontrolled ROS production, suggesting potential therapeutic benefits for various cardiovascular conditions, as shown in Figure 6 [9,80,84].

Hypertrophic cardiomyopathy is associated with disrupted cardiac redox balance in genotype-positive patients, even without genetic alterations [56,84,85,86]. Factors such as wall stress, mitochondrial or microvascular dysfunction, and asymmetric septal remodeling can trigger this imbalance alongside age-related changes in cellular redox balance. Diet, antioxidants, and genetic predisposition also influence cardiac redox status. Genetic polymorphisms can affect phenotypic expression in nonfamilial HCM. Understanding the importance of these factors in other pathological mechanisms, such as myocardial energy depletion, requires further research. Current HCM therapies include β-blockers, Ca^2+^ antagonists, ACE inhibitors, diuretics, antiarrhythmics, and anticoagulants, potentially impacting cardiac redox status [29].

SS-31, also known as Elamipretide, a cell-permeable peptide, selectively targets the inner mitochondrial membrane via cardiolipin binding, thereby reducing mitochondrial ROS production and restoring mitochondrial bioenergetics [87]. It stabilizes cardiolipin, promotes membrane restructuring, and facilitates the formation of mitochondrial supercomplexes, thereby reducing ROS production [87,88,89,90]. Recent studies have focused on mitochondrial targets such as elamipretide and NAD+ supplementation. In HCM myectomized tissues, elamipretide treatment increased complex I incorporation and enhanced NADH-binding respiration. NADH-linked breathing impairment in HCM may stem from inefficient coupling of complex I to complexes III and IV, reduced Krebs cycle dehydrogenases, and inadequate mitochondrial calcium uptake [23,81].

Mitochondria-focused antioxidant treatments are promising for treating conditions such as HCM [91,92,93]. Recent studies have demonstrated the mitigating effects of these antioxidants on metabolic disorders. Traditional vitamins and antioxidant compounds such as coenzyme Q, alpha-lipoic acid, and N-acetylcysteine can combat excessive ROS production. Two mitochondrion-focused antioxidants, MitoQ, and MitoVit E, have become prominent. MitoQ targets mitochondria by incorporating the triphenylphosphonium lipophilic cation (TPP) group into coenzyme Q10 ubiquinone, reducing ROS and protecting against age-related damage. MitoVitE, a derivative of vitamin E conjugated with TPP, also targets mitochondria [9]. Coenzyme Q (CoQ) improves symptoms and quality of life in patients with HCM and reduces the interventricular septum thickness. It acts as an electron transporter in the inner mitochondrial membrane, restoring redox balance and preventing mitochondrial ROS formation. It also reduces left ventricular outflow obstruction and ventricular tachycardia [94,95].

Chen et al. (2015) discovered that 17b-estradiol improved myocardial diastolic function in HCM mice, suggesting that estrogen protects against disease progression [56]. The intervention reduced oxidative stress and prevented myocardial dysregulation.

1-Deoxynojirimycin, a treatment for morphological mitochondrial disorders, has shown benefits in recovering mitochondrial cristae and improving calcium homeostasis. Its action on optic atrophy protein 1 (OPA1), a regulator of mitochondrial cristae formation, enhances the number and width of the mitochondrial cristae, thereby improving ATP production and calcium homeostasis. Thus, 1-deoxynojirimycin is a potential therapeutic agent for treating HCM [41].

Regarding metabolic alterations, a study in rats highlighted the beneficial effects of restoring fatty acid metabolism on cardiac hypertrophy. Treatment with Tricaprylin reduced cardiomyocyte cross-sectional area, decreased interstitial fibrosis, and lowered biomarkers related to oxidative stress [96]. However, it is important to note that metabolic drugs like Perhexiline can also offer protective effects by shifting fatty acid metabolism to carbohydrates. This metabolic shift increases ATP supply and protects against catecholamine-induced cardiac damage. While fatty acid metabolism plays a crucial role in cardiac energetics, balancing its regulation with other metabolic pathways, such as carbohydrate metabolism, may be essential for optimizing cardiac function in conditions like HCM. In this regard, in HCM models, Perhexiline improved phenotypic characteristics and exercise capacity in symptomatic patients. However, chronic shift to glycolytic metabolism may be detrimental, indicating that intermittent metabolic therapy may be a more effective and innovative approach, especially compared to continuous treatment. In this context, a phase 2 clinical trial with Perhexiline showed some modest improvements in symptoms and cardiac bioenergetics compared to placebo, but the drug was discontinued due to multiorgan toxicity [97,98]. Other drugs inhibiting fatty acid β-oxidation also indicated negative outcomes, such as trimetazidine, an inhibitor of 3-ketoyl-CoA thiolase [99]. Given the negative regulation of enzymes involved in fatty acid oxidation in HCM patients, a more promising therapeutic approach would be to increase the availability of alternative fuel sources for ATP production. Therapeutic ketosis, which increases ketone bodies, has shown to be a viable strategy, considering cardiac metabolism adaptation in HCM patients. β-hydroxybutyrate, a ketone body, increased in the hearts of HCM patients, suggesting possible adaptation to reduced availability of acyl carnitines. Pharmacological oxidation of ketone bodies may have potential therapeutic applications [100,101,102].

B3-adrenergic receptors (b3AR) offer potential treatment avenues for metabolic changes in HCM. Activation of b3AR positively affects intracellular signaling, enhances calcium uptake, improves diastolic relaxation, and exhibits antioxidant properties [103].

Xanthine oxidase (XO) significantly contributes to ROS and uric acid generation, which are byproducts of purine metabolism. Elevated uric acid levels in patients with heart failure and HCM are correlated with an increased risk of cardiovascular events. Future research should confirm the activity of XO in HCM and investigate the blocking of ROS production by XO as a therapeutic strategy [104,105]. Restoring fatty acid metabolism in cardiac hypertrophy has beneficial effects, including reducing the cardiomyocyte cross-sectional area and oxidative stress biomarkers [96]. Perhexylin, a metabolic drug, switches metabolism from fatty acids to carbohydrates, boosting ATP delivery and protecting against catecholamine-induced cardiac damage. Intermittent metabolic therapy may be more beneficial than continuous treatment [97,98,99]. Therapeutic ketosis, which involves elevated levels of ketone bodies, is a viable strategy for cardiac metabolism adaptation in patients with HCM [100,101,102].

#### 3.2.2. Emerging Drug Therapies for HCM

Omecamtiv mecarbil (OM) is a myosin activator that has been studied as a potential treatment for systolic heart failure. Its effects may vary depending on the intracellular calcium concentration, which suggests that it could enhance contractility in cardiac failure and reduce it in diastolic failure, such as in HCM. MyoKardia, Inc. (Brisbane, CA, USA) has developed a novel oral medication called Mavacamten, which acts as an allosteric modulator of cardiac β-myosin [34]. This drug can inhibit the actin-myosin cross-bridge, reducing contractility and oxygen consumption in HCM models [106,107,108]. By weakening the bond between myosin heads and actin fibers, Mavacamten can prevent cardiac hypertrophy and fibrosis. It may also indirectly affect mitochondrial and energetic dysfunction by modulating contractility and cardiac function, optimizing energy balance in the heart, and enhancing efficiency in energy utilization [109]. It is worth noting that while Mavacamten shows promise as a treatment for HCM, Omecamtiv mecarbil, designed for systolic heart failure, would not be effective for HCM due to its mode of action.

Verapamil is a drug that blocks L-type calcium channels and has shown promising results in reducing hypertrophic cardiomyopathy in mice with the MYH7-Arg403Gln mutation. However, its effectiveness may vary in patients with multiple mutations. Understanding the early mechanisms involved in this disease is important to develop effective treatments. Genetic modifiers, epigenetic variations, and environmental factors can all play a role in the development of HCM, leading to various clinical outcomes [110].

The VANISH clinical trial used valsartan, an angiotensin receptor antagonist, to treat HCM with sarcomeric mutations. However, it did not significantly impact slowing subclinical HCM progression [111,112]. Sacubitril/valsartan, an angiotensin II receptor inhibitor/antagonist, has been tested in a clinical trial of patients with HCM. Lifestyle interventions such as regular exercise can activate beneficial metabolic pathways [113]. There is an unmet need for the preventive treatment of HCM. A murine model study assessed the efficacy of in vivo treatment using a derived peptide (AID-TAT) to restore mitochondrial metabolic activity and prevent HCM with the cardiac mutation Gly203Ser in troponin I [114,115], as shown in Table 1.

## 4. Conclusions

Hypertrophic cardiomyopathy (HCM) represents a complex cardiac condition marked by a spectrum of biological process alterations, including ionic homeostasis, structural remodeling, metabolic balance, and ROS regulation. Its current therapeutic landscape primarily targets symptom management and the prevention of complications, highlighting a significant gap in the availability of more effective treatment methodologies. The treatment options currently range from symptom alleviation strategies to more invasive surgical interventions, with the choice of treatment tailored to the evolving profiles of individual patients.

A key focus of researchers and clinicians is the investigation of modifiable risk factors encompassing genetic predispositions and mutations. This approach underscores the recognition that HCM is not just a product of unchangeable genetic factors but may also be influenced by modifiable elements, thus opening new avenues for intervention and management.

A comprehensive understanding of the biomolecular, genetic, and mitochondrial mechanisms that drive HCM pathophysiology is vital. This knowledge is crucial for the advancement of research in this field and the development of new and more effective therapeutic modalities. As our understanding deepens, it paves the way for innovative treatments that move beyond symptom management, potentially offering significant and long-lasting benefits to patients with this condition.

In summary, HCM presents a multifaceted challenge that requires a multifaceted response, encompassing advanced research, innovative therapeutic strategies, and a personalized approach to patient care. The future of HCM treatment lies at the intersection of these diverse yet interconnected domains.

## Figures and Tables

**Figure 1 ijms-25-05855-f001:**
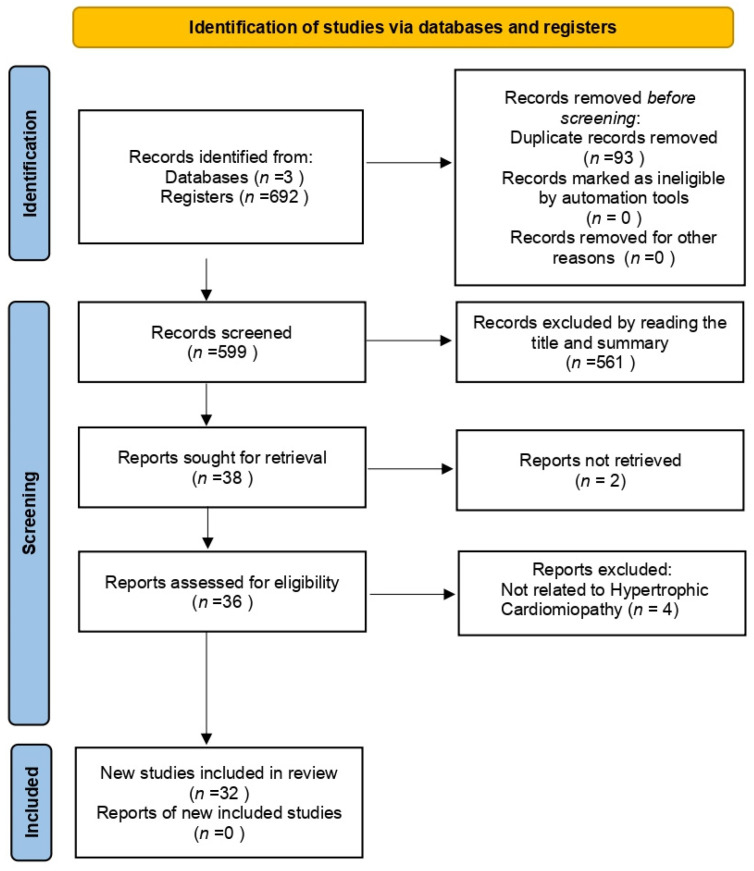
PRISMA flowchart of the reviewed articles.

**Figure 3 ijms-25-05855-f003:**
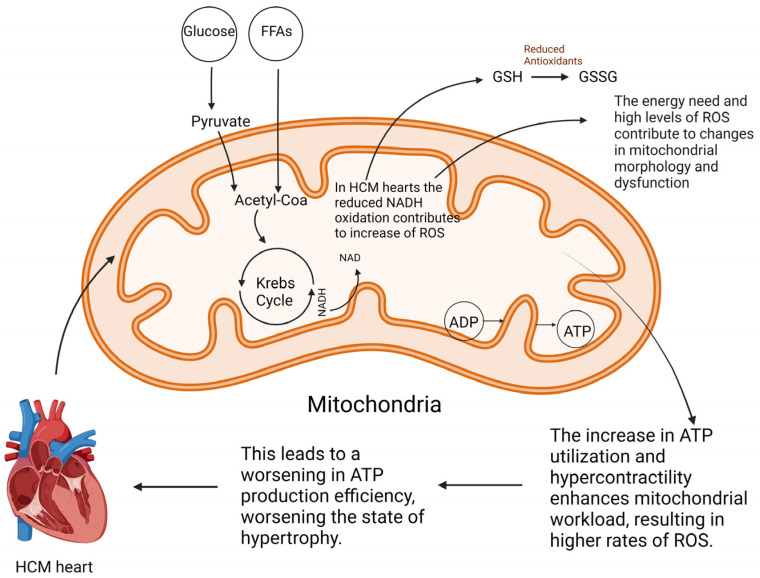
Metabolic redox integration in cardiomyocytes. Legends: HCM = Hypertrophic Cardiomyopathy, FFAs = Free Fatty Acids, NAD = Nicotinamide Adenine Dinucleotide, NADH = Nicotinamide Adenine Dinucleotide Hidrogenado, ADP = Adenosine Diphosphate, ATP = Adenosine Triphosphate, GSH = Glutathione, GSSG = Oxidized Glutathione, ROS = Reactive Oxygen Species.

**Figure 2 ijms-25-05855-f002:**
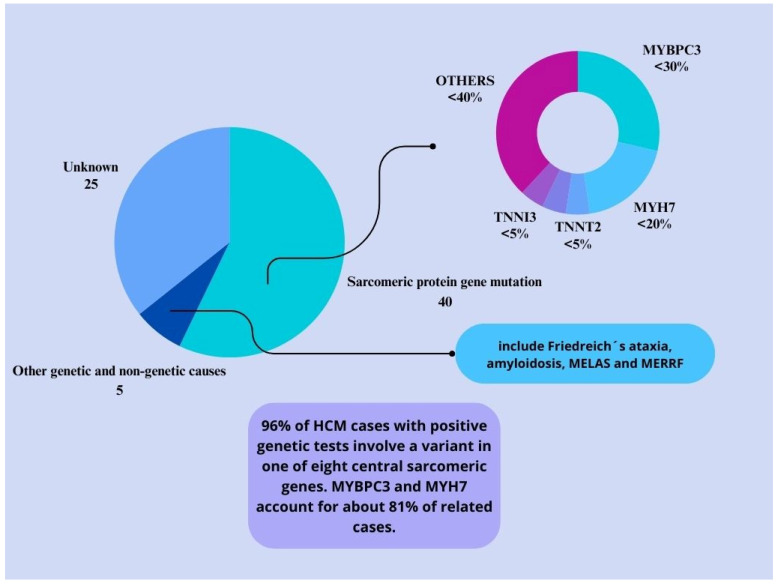
Genetic and non-genetic causes of hypertrophic cardiomyopathy. Legends: MELAS = mitochondrial encephalomyopathy, lactic acidosis, and stroke-like episodes; MERRF = myoclonic epilepsy with ragged red fibers; MYBPC3 = myosin-binding protein C, cardiac-type; MYH7 = myosin, heavy chain 7; TNNI3 = troponin I, cardiac; TNNT2 = troponin T, cardiac.

**Figure 4 ijms-25-05855-f004:**
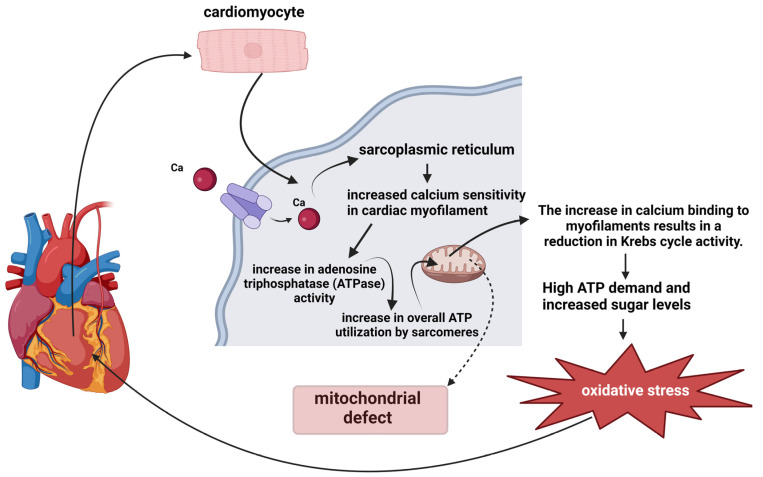
Metabolic fluxes under high ATP demand in hypertrophic cardiomyopathy. Legends: ATP = adenosine triphosphate, and ATPase = enzyme.

**Figure 5 ijms-25-05855-f005:**
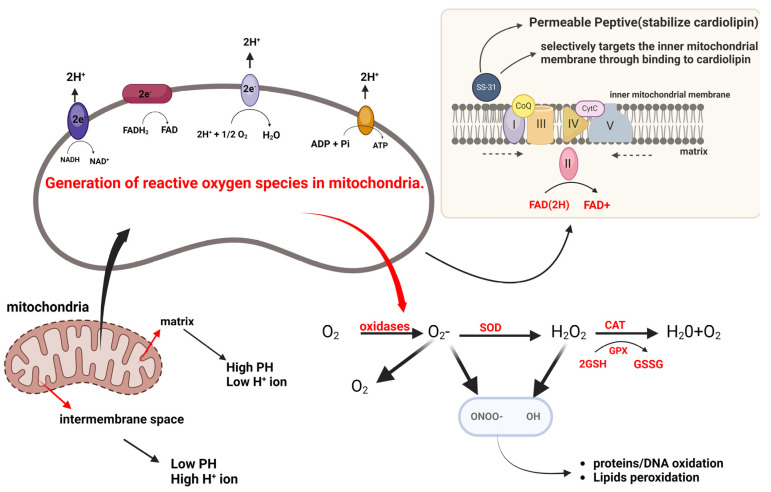
Mitochondrial electron transport chain and ROS generation. Legends: NADH = adenine dinucleotide; NAD⁺ = nicotinamide adenine dinucleotide; PH = potential of hydrogen; SOD = superoxide dismutase; CAT = catalase; GSH = glutathione; GSSG = oxidized glutathione; ADP = adenosine diphosphate; ATP = adenosine triphosphate; Complex I = NADH dehydrogenase; Complex II = Succinate dehydrogenase; Complex III = Cytochrome b-c1; Complex IV = Cytochrome oxidase; Complex V = ATP synthase.

**Figure 6 ijms-25-05855-f006:**
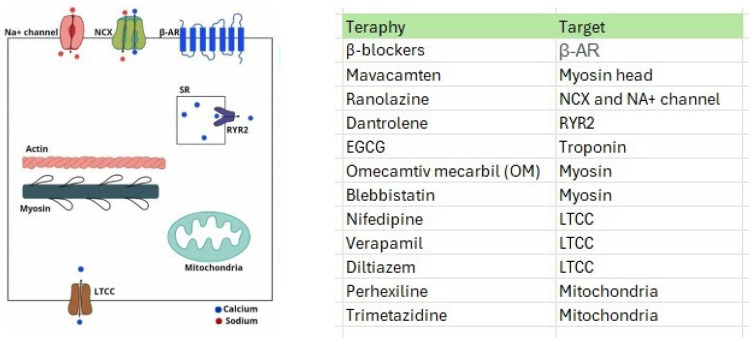
Ongoing pharmacological strategies to treat HCM. Legends: EGCG = epigallocatechin-3-gallate, LTCC = L-type calcium channel, NCX = Sodium-Calcium Exchanger, β-AR = beta-adrenergic receptor, RYR2 = ryanodine receptor type 2, SR = sarcoplasmic reticulum.

**Table 1 ijms-25-05855-t001:** Therapeutic possibilities and their primary effects.

Therapeutic Possibilities	References	Primary Effects
Alpha-lipoic acid	[9]	Neutralizes free radicals, reduces oxidative stress, improves endothelial function, increases nitric oxide production, and influences cellular signaling pathways associated with cardiovascular health.
N-acetylcysteine	[9,116,117]	Has mucolytic and fluidizing activity. Reduces oxidative damage to heart tissues and increases the production of nitric oxide, which helps dilate veins and improve blood flow.
MitoQ	[9]	Neutralizes free radicals within the mitochondria, reduces oxidative stress, and preserves mitochondrial function.
MitoVit E	[9]	Has antioxidant capacity aimed at protecting mitochondria against damage caused by free radicals and reducing oxidative stress.
Coenzyme Q	[9,94,95]	CoQ is involved in the mitochondrial electron transport chain, where it aids in generating adenosine triphosphate (ATP), the cellular energy currency. Additionally, CoQ possesses antioxidant properties, shielding the mitochondria and the heart against damage caused by free radicals during the energy production process.
Elamipretide/SS-31	[87,88,89,90]	It concentrates on mitochondrial membranes and helps regulate energy production by enhancing the efficiency of the electron transport chain. Additionally, Elamipretide can reduce oxidative stress, stabilize mitochondrial membranes, and preserve mitochondrial integrity.
17b-estradiol	[56]	It influences cardiac function by regulating genes, interacting with specific receptors, modulating vascular response, and indirectly affecting metabolic and cellular signaling processes.
1-Deoxynojirimycin	[41]	Inhibits metabolic pathways involved in protein glycosylation.
Tricaprylin	[96]	It acts as a rapid source of fuel that can be used for ATP production, thereby aiding in the heart’s contractile function. This can be beneficial, especially in situations where the heart has an increased energy demand, such as certain cardiac conditions or intense physical exercise.
Perhexiline	[97,98,118]	Inhibits cardiac fatty acid uptake, promoting glucose utilization and reducing oxygen demand.
Trimetazidine	[99]	Blocks the mitochondrial protein responsible for transporting fatty acids into mitochondria. This action favors glucose metabolism, preserving cardiac function during low-oxygen-supply conditions such as angina. It improves cardiac efficiency and reduces angina pain.
Beta-3 adrenergic receptors	[103]	It is primarily activated by neurotransmitters like norepinephrine and adrenaline. It is involved in regulating fat metabolism and thermogenesis (heat production) and can influence cardiac function, lipolysis (fat breakdown), and muscle contraction in specific tissues, such as the smooth muscle of the bladder.
Omecamtiv Mecarbil	[106]	A myocardial activator that increases the sensitivity of the cardiac muscle (myocardium) to calcium concentration, resulting in greater efficiency in cardiac muscle contraction. Binds to myosin in the myocardium, altering its conformation and allowing a more effective interaction with calcium.
Mavacamten	[34,106,107,108]	An allosteric inhibitor of heart-specific myosin adenosine triphosphatase.
Diltiazem	[110]	A non-selective calcium channel antagonist that affects the heart and blood vessels.
Valsartan	[111,112]	Displaces angiotensin II from the AT1 receptor and produces its blood-pressure-lowering effects by antagonizing AT1-induced vasoconstriction, aldosterone release, catecholamine release, arginine vasopressin release, water intake, and hypertrophic responses.
Sacubitril	[119]	Simultaneously inhibits neprilysin (neutral endopeptidase; NEP) through sacubitrilat, the active metabolite of the prodrug sacubitril.
AID-TAT	[114,115]	This is related to activating the enzyme AMPK (AMP-activated protein kinase). This leads to a series of cellular events, including increased glucose uptake and increased ATP production.

## Data Availability

The original contributions presented in the study are included in the article and Appendix A; further inquiries can be directed to the corresponding author.

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
