# Peer review of "Genetic Mutations and Mitochondrial Redox Signaling as Modulating Factors in Hypertrophic Cardiomyopathy: A Scoping Review"

_ijms, 2024, doi:10.3390/ijms25115855_

Round 1

Reviewer 1 Report

Comments and Suggestions for Authors

This manuscript is a scoping review of the relationship between HCM mutations and mitochondrial redox signalling. My major concern is with the methodology that is not adequately described.  

-What search terms are used for the initial identification of papers?

-Were any papers that the authors knew about but were not retrieved in the initial search included or was the study entirely in silico?

-of 599 papers retrieved  32 were selected in a screening process.  Was this based on the authors study of the papers or was screening based on a machine learning algorithm?  In which case we must be shown what algorithm was used and the criteria for screening.

-It is noted that the authors recently published a scoping review on a similar subject (Atrial fibrillation and mitochondrial dysfunction). 

  • DOI: 
  • 10.3390/ijms25010535

 The methods were slightly better described here but were still inadequate.

-  The summaries of findings are indisputable and clear, however there is really no analysis of these results that would add value to the review on a subject that has been reviewed before.   There is also a certain lack of perspective.  For instance Omecamtiv Mecarbil and Mavacamtem have precisely opposite effects yet they appear in the same paragraph with no comment that mavacamten is a very useful treatment for HCM whilst Omecamtiv is intended treat heart failure and would be useless for HCM.

Another important point could be extracted from your data:  is mitochondrial dysfunction due to the direct effects of the mutation (e.g. hyper contractility)  or is it due to the hypertrophy and its secondary effects.

-There are several mistakes in figure 6:  EGCG interacts with troponin.  Omecamtiv and  blebbistatin interact with myosin.

Author Response

Dear Reviewer 1,

Thank you for your insightful feedback on our manuscript titled "Genetic Mutations and Mitochondrial Redox Signaling as Modulating Factors in Hypertrophic Cardiomyopathy: A Scoping Review." We greatly appreciate the constructive criticisms that have been crucial in enhancing the academic rigor of our work. In response to your comments, we have made the following revisions to improve the clarity and coherence of our article:

Major Points:

  1. Paragraph Reorganization:
    • We acknowledge the need for clearer delineation between the topics of genetic mutations and mitochondrial redox signaling as related to hypertrophic cardiomyopathy (HCM). Accordingly, we have restructured our manuscript by creating two distinct main sections, each further divided into detailed subsections. This new organization facilitates a systematic and thorough discussion on the types of mutations, therapeutic approaches, and the role of mitochondrial redox signaling.
  2. Clarification of Citations:
    • We have updated Supplementary Table 1 to include clear citations for the 32 articles directly referenced in our review, ensuring greater transparency and distinction from other sources used in our manuscript.

Minor Points:

  1. Figure 1:
    • We have corrected the PRISMA flowchart to align with standard guidelines.
  2. Textual Clarity:
    • We have resolved the grammatical issue on page 5, line 236, by inserting an appropriate verb, thus enhancing the sentence structure and readability.
  3. Italicization:
    • The term "in vivo" has been italicized throughout the document, ensuring consistency with scientific formatting standards.
  4. Technical Corrections:
    • Accurate notation has been applied to "Ca2+" to maintain consistency and precision in our scientific communication.
  5. Figure Captions:
    • Following your recommendations, we have revised all figure captions to include only abbreviations within the legends, with explanatory text moved to the main body of the captions.
  6. Consistency in References:
    • We have thoroughly reviewed and rectified any inconsistencies in the referencing style between the supplementary materials and the main text to ensure uniformity and accuracy.
  7. Supplementary Table 1:
    • We have added citation numbers to the entries in Supplementary Table 1 to aid in cross-referencing within the main article, as suggested.

We trust these revisions address your concerns effectively and enhance the manuscript's alignment with the standards of the International Journal of Molecular Sciences. We are committed to continuous improvement of our work and are grateful for the opportunity to contribute meaningfully to the field of cardiology.

Thank you for considering our manuscript for publication. We look forward to your response.

Sincerely,

Antônio da Silva Menezes Jr.
Faculty of Medicine, Department of Clinical Medicine,
Federal University of Goiás (UFG)
a.menezes.junior@uol.com.br

Reviewer 2 Report

Comments and Suggestions for Authors

In this review the authors addressed specific modulating factors involved in hyperthrophic cardiomyopathy such as genetic mutations and mitochondrial redox signalling. Authors in particular selected some papers and focused attention on different aspects related to the modulation factors involved in HCM. In this sense, in my opinion the work is not well constructed because there is no a clear presentation and division in the different paragraphs of the modulating factors (genetic mutations and mitochondrial redox signalling) and of the main papers selected. Major and Minor points are reported in the attached file.

Author Response

Dear Reviewer 2,

Thank you for your insightful comments and constructive criticisms regarding our manuscript, "Genetic Mutations and Mitochondrial Redox Signaling as Modulating Factors in Hypertrophic Cardiomyopathy: A Scoping Review." We have thoroughly reviewed each of your points and have made the following revisions to our manuscript:

  1. Search Terms and Methodology:
    • We employed a combination of search terms related to hypertrophic cardiomyopathy (HCM), genetic mutations, and mitochondrial redox signaling to ensure a comprehensive retrieval of relevant literature. These terms were chosen for their direct relevance to our review's scope.
    • We adhered to the scoping review methodology outlined by the Preferred Reporting Items for Systematic Reviews and Meta-Analyses Extension for Scoping Reviews (PRISMA-ScR).
    • The article screening was conducted manually. We have now expanded the Methods section to include more detailed criteria for article selection.
  2. Comparison with Previous Scoping Reviews:
    • We acknowledge the previous review on atrial fibrillation and mitochondrial dysfunction, cited as DOI: 10.3390/ijms25010535. We have refined our methodology description, drawing on the lessons learned from previous works.
  3. Analysis of Results and Therapeutic Perspectives:
    • A more detailed discussion on the therapeutic implications of treatments such as Omecamtiv Mecarbil and Mavacamten has been added. This includes an examination of their distinct mechanisms of action and potential benefits in HCM treatment.
  4. Mitochondrial Dysfunction and Mutation Effects:
    • We appreciate your suggestion to distinguish between mitochondrial dysfunction arising directly from mutations and that secondary to hypertrophy. We have clarified that mitochondrial dysfunction can be a primary consequence of genetic mutations or a secondary outcome of pathological hypertrophy, each with distinct bioenergetic consequences.
  5. Corrections in figure 6:
    • We apologize for the inaccuracies previously present in figure 6. The figure has been corrected to accurately depict the interactions between EGCG, Omecamtiv, and Blebbistatin with their respective targets.

We believe these revisions have substantially improved the clarity and quality of our manuscript, addressing your concerns comprehensively. We are grateful for the opportunity to enhance our submission and thank you for your valuable feedback.

Sincerely,

Antônio da Silva Menezes Jr.
Faculty of Medicine, Department of Clinical Medicine,
Federal University of Goiás (UFG)
a.menezes.junior@uol.com.br

Reviewer 3 Report

Comments and Suggestions for Authors

Congratulations to the authors for taking up the interesting topic of HCM. The authors searched an extensive database of scientific data, almost 120 literature items. Plus for mentioning mavacamten, a new drug that may be a chance for patients with HCM. The article is supplemented with important figures and tables. The genetic aspects of HCM have been extensively discussed.

Author Response

We would like to express our sincere appreciation for your thoughtfulness.

Round 2

Reviewer 1 Report

Comments and Suggestions for Authors

A good revision

Reviewer 2 Report

Comments and Suggestions for Authors

Thanks to the authors that have addressed properly the suggestions that made better this version of the paper, now suitable for publication.